# Population dynamics of western gorillas at Mbeli Bai

**Andrew M. Robbins[1‡]\*, Marie L. Manguette[1,2‡], Thomas Breuer[2,3], Milou Groenenberg[2], Richard J. Parnell[4], Claudia Stephan[1,2,3], Emma J. Stokes[4], Martha M. Robbins[1]**

**1** Wildlife Conservation Society–Congo Program, Brazzaville, Republic of Congo, **2** Nouabalé-Ndoki Foundation, Brazzaville, Republic of Congo, **3** Division of Developmental Biology, Friedrich-Alexander University Erlangen, Erlangen, Germany, **4** Wildlife Conservation Society, Global Conservation Program, Bronx, NY, United States of America

‡ AMR and MLM are share first authorship on this work.
\* htram@aol.com

**Data Availability Statement:** The archive data is in a Supporting Information file.

**Funding:** The author(s) received no specific funding for this work.

## Abstract

Long-term studies of population dynamics can provide insights into life history theory, population ecology, socioecology, conservation biology and wildlife management. Here we examine 25 years of population dynamics of western gorillas at Mbeli Bai, a swampy forest clearing in Nouabalé-Ndoki National Park, the Republic of Congo. The Mbeli population more than doubled from 101 to 226 gorillas during the study. After adjusting for a net influx of gorillas into the study population, the increase represents an inherent growth rate of 0.7% per year, with 95% confidence limits between -0.7% and 2.6%. The influx of gorillas mainly involved immigration of individuals into existing study groups (social dispersal), but it also included the appearance of a few previously unknown groups (locational dispersal). The average group size did not change significantly during the study, which is consistent with the possibility that western gorillas face socioecological constraints on group size, even when the population is increasing. We found no significant evidence of density dependence on female reproductive success or male mating competition. The distribution of gorillas among age/sex categories also did not change significantly, which suggests that the population had a stable age structure. Our results provide evidence of population stability or growth for some western gorillas (albeit within a small area). The results highlight the value of law enforcement, long-term monitoring, and protected areas; but they do not diminish the importance of improving conservation for this critically endangered species.

## Introduction

Along with other taxa, the order Primates is facing an extinction crisis due to habitat destruction, poaching and illegal activities, disease, and climate change [1–3]. To combat this crisis, a key component to conservation efforts is the monitoring of population dynamics [4, 5]. In addition to human disturbances, changes in population size and structure can also arise from natural environmental shifts, density dependence, as well as demographic stochasticity [6–8].

**Competing interests:** The authors have declared that no competing interests exist.

On a local scale, the proximate causes of variation in population dynamics can include changes in the rates of births and deaths, as well as the influx and outflow of individuals [9, 10].

Studies of population dynamics can contribute to conservation management decisions by identifying local regions with more births than deaths and a net outflow of individuals [11, 12]. Such regions are known as a "source", and are considered priorities for supporting broader populations [13, 14]. In contrast, a "sink" is an area with fewer births than deaths, and a net influx of individuals to sustain its population. Other regions with a net influx are considered a "refuge", if they have higher population density than the surrounding area, which has been found near long-term research stations [15, 16].

One of the most common topics in studies of population dynamics is the density dependence of demographic rates [17–19]. An increase in population density is predicted to affect demographic parameters in the following sequence: higher offspring mortality, followed by a later age of first reproduction, lower female fertility, and ultimately higher adult mortality [20, 21]. If females are not annual breeders, however, then fertility could decline before offspring mortality [22]. When direct measurements of those demographic parameters are not available, they are sometimes estimated from the age structure to assess the population dynamics [23, 24]. For example, a growing population will often have a younger age distribution than a declining population [5, 25].

In addition to its effect on demographic parameters, changes in population density can also influence the social organization of species that live in groups [26–28]. A change in population density may lead to an increase or decrease in number of groups and/or group size. For example, increased feeding competition may limit the size of groups as a population grows [29, 30]. Increased infanticide may limit population growth as male mating competition intensifies [27, 31]. Disease and anthropogenetic threats may disproportionately affect particular age/sex classes that can impact the social grouping patterns [5, 32, 33]. In sum, assessing the viability of social species requires monitoring of not merely the absolute numbers of individuals, but also changes in group size, dispersal, and the age/sex class structure [34]. More information is needed to understand the population dynamics of primates, including the inter-relationships among socioecology, behavioral ecology, life histories, and conservation.

The objective of this study is to examine the population dynamics of western gorillas over 25 years at Mbeli Bai, a swampy forest clearing within the Nouabalé-Ndoki National Park (NNNP) in the Republic of Congo. Western gorillas (*Gorilla gorilla gorilla*) are critically endangered despite their wide distribution across Central Africa [35]. The total population is estimated to be 330,000 western gorillas, which has been declining by 2.7% per year due to illegal killings, disease, and habitat degradation [36]. For example, outbreaks of the Ebola virus have been blamed for 80-95% losses of two western gorilla populations within the Odzala-Kokoua National Park in the Republic of Congo [37–39]. In contrast, western gorilla populations have been stable in the NNNP and nearby areas [36, 40]. However, the NNNP has a lower gorilla density than some of the nearby areas, which has been attributed to lower habitat quality rather than differences in conservation strategies [41, 42]. Mbeli suffered from high levels of poaching before our study began, but no poaching of gorillas has been reported since then [42–46]. We expect that the presence of the research site and park protection could lead to a more stable or increasing population during this study.

Western gorillas typically live within groups that contain one adult male (silverback), a few adult females, and their immature offspring [47–49]. Immature gorillas leave the group before reaching adulthood. Males emigrate to become solitary, whereas females transfer directly to another group or to a solitary male [45, 50]. Females may transfer repeatedly throughout adulthood, but "voluntary" dispersal generally does not occur when they have dependent offspring, who could be vulnerable to infanticide by other silverbacks [51, 52]. Groups disintegrate when

the silverback dies: the remaining gorillas join other silverbacks ("involuntary" dispersal), except for the infants who usually disappear due to suspected infanticide [51, 53]. Infanticide can also occur during encounters between groups when the silverbacks are still alive [27, 53–55].

Previous studies at Mbeli and other bais have provided information on the social structure, reproductive success and dispersal of western gorillas [45, 47, 51, 56]. Studies of population dynamics of western gorillas have mainly focused on population recovery from an Ebola outbreak at Lokoué Bai in the Republic of Congo, where the population remained 85% smaller than pre-Ebola levels, despite an annual increase of 3.5% in the subsequent ten years [39, 57]. Among healthy populations of western gorillas, density dependence has not been reported for female reproductive success or male mating competition. Among mountain gorillas, interbirth intervals were significantly longer and age of weaning was later at Bwindi versus the Virungas, which was considered potential evidence that the Bwindi population might be closer to its carrying capacity, or due to differences in ecological conditions [22, 58]. Density dependence in male mating competition has been suggested for a subsection of the Virunga mountain gorilla population, where an increase in group density coincided with seven deaths of adult males, and a five-fold increase in infanticide accounted for more than half of the reduction in population growth [27]. Western gorillas may face socioecological constraints on group size based on feeding competition and/or male mating competition, because females were more likely to leave larger groups for smaller ones [45]. Evidence for those socioecological constraints has been weaker for mountain gorillas [59, 60]. A high proportion of immature gorillas in census surveys (>40%) has been considered evidence of a stable or growing population for mountain gorillas, but the metric has not been tested for western gorillas [61–63].

To examine the population dynamics at Mbeli, we first calculated an annual rate of population increase, as well as an inherent growth rate that adjusts for the influx and outflow of gorillas between the study area and the surrounding region. We used those results to determine whether Mbeli is a source, a sink, or a refuge. Second, to look for potential effects of density dependence upon female reproductive success, we analyzed whether the population size was correlated with infant mortality, the age of first birth, interbirth intervals, birth rates, adult mortality and/or dispersal. If the gorilla density is limited by habitat quality (relative to nearby areas), then we expect female reproductive success to decline and emigration to increase when the population grows. Third, to look for potential effects of density dependence upon male mating competition, we analyzed whether the number of groups was correlated with dominant male mortality, and with suspected infanticide when the dominant male died. Fourth, to check for socioecological constraints upon social structure, we examined whether changes in the population size were more strongly correlated with the number of groups or with group size. Finally, to examine whether the population has a stable age structure, we looked at the percentage of immatures and the adult sex ratio. We compare our results with other primate populations including mountain gorillas.

## Methods

### Study site and population

Western gorillas were studied at a 12.9ha swampy forest clearing (known as a "bai") in the Nouabalé-Ndoki National Park, in the Republic of Congo [45, 47, 51, 56]. The gorillas were observed from a 9m high platform which provides almost complete visibility of the bai [64]. The bai was monitored from February 1995 through April 2020, except for six-month interruptions in 1997 and 2016. After excluding those interruptions, the 77,176 hours of observations represent an average of 9.0 hours per day during an average of 27.9 days per month (Inter-Quartile Ranges = 8.4–9.8 hours per day, and 27–31 days per month).

Gorillas were identified based on their body size, hair coloration, and facial features such as the shape of the brow-ridge and ears and nose prints [65, 66]. Each group does not visit the bai every day, so we used an initialization period to define the starting population. Thirteen groups were identified from February through December 1995, with an average of 21 observed visits per group (Inter-Quartile Range = 5–36 visits per group). Only one additional group was identified during the next three years. The additional group contained only a silverback and two adult females with no offspring, which suggests that it may have formed recently, rather than being missed during the first year [64]. Therefore, we treated 1995 as the initialization period and our analyses use the data from 1996 onward. The additional group (and the subsequent appearances of other gorillas) was considered influx, rather than part of the starting population.

Due to the intermittent bai visits, the dates of birth, dispersal, and disappearances were typically estimated. Some gorillas have been observed within 1-2 days after their birth, which is known because their group had just been seen without them. When gorillas were first observed beyond that age, their birthdates were estimated by comparing their morphology and behavioral characteristics with other gorillas whose age was already determined [47, 65, 67]. The precision of those birthdates is estimated to be within a few weeks for most gorillas who were first observed as infants, up to 1-2 years for gorillas who were first observed as they approached adulthood, and ± 4-10 years for adults.

Gorillas of both sexes were classified infants until age four, and then juveniles until age 7.5 [65]. Females were classified as subadults from ages 7.5–10, and adults from age 10 onward [65]. Males were classified as subadults from ages 7.5–11, blackbacks from ages 11–14, young silverbacks from 14-18, and mature silverbacks from age 18 onward. Infants, juveniles, and subadults are considered immature gorillas [63]. The mature silverback in a group is classified as "dominant" over the other gorillas who are "subordinates". "Breeding groups" contain a mature silverback and at least one adult female, whereas "nonbreeding groups" do not contain adult females. The term "social unit" includes either type of group as well as solitary males. The appearance of a new social unit was defined as "locational dispersal" into the study area, and the immigration of a gorilla into a known social unit was defined as "social dispersal" from one social unit into another.

When gorillas were observed in the same group during two consecutive visits to the bai, we assumed that they had stayed with the group between those two visits. When a gorilla changed groups, we typically assumed that the gorilla had transferred directly from the source group to the destination group, and we estimated the dispersal date as the midpoint between the last observed visit of the source group and the first observed visit of the destination [50, 51]. However, if the destination group was observed without the gorilla, even after the source group had already been observed without it, then we defined the temporary absence as an intermediate dispersal to an unknown destination. The dates of temporary and permanent disappearances were estimated as the midpoint between the last date that the gorilla was observed, and the first date that the source group was observed without the gorilla.

When gorillas disappeared permanently (i.e., they were never observed again), we could not determine whether they had dispersed out of the study population (outflow) or died. When a dominant male disappeared, we assumed that he had died because such males do not emigrate, and in most cases, the male had become seriously injured or very thin [50]. Infanticide is expected after the dominant male dies, so we assumed that infants had died when they disappeared during the subsequent group disintegration. Voluntary dispersal begins at age 7.5 for females and age 12 for males, so when gorillas disappeared from a stable group before those ages, we assumed that they had died (Supporting Information). Hypothetically, those assumed deaths could be due to predation, disease, infanticide, or other causes. All other cases

of missing gorillas were classified as unexplained disappearances (which could be either deaths or outflow).

## Population growth

We calculated the "annual rate of population increase" using Eq 1, where $N_i$ is the initial number of gorillas (as of January 1, 1996), $N_f$ is the final number of gorillas, and y is the duration of the study in years [63].

$$\text{Rate of increase} = [(N_f/N_i)^{1/y}] - 1 \qquad (1)$$

Eq 1 does not adjust for the influx or outflow of gorillas during the study, so it could be sensitive to our definition of the starting population versus subsequent influx. To adjust for such assumptions, we calculated a "growth rate" with a time-series calculation that accounts for influx and outflow [30, 68, 69]. The growth rate calculations began with the initial number of gorillas, and then used Eq 2 to estimate the number of gorillas in each subsequent month:

$$N_i = [N_{i-1} * (1 + r_m)] + A_i \qquad (2)$$

where $N_i$ is the number of gorillas in month $i$, $N_{i-1}$ is the number of gorillas in the previous month, $r_m$ is the monthly growth rate. The adjustment factor $A_i$ equals the influx of gorillas from the surrounding population into the study area, minus the outflow of gorillas from the study area into the surrounding population.

Eq 2 is analogous to calculations for the balance of a savings account, in which the adjustments ($A_i$) are similar to deposits and withdrawals, and $r_m$ is similar to an interest rate that is compounded monthly. We used monthly increments because the final value depends on when the adjustments occurred (e.g., the final balance of a savings account will depend on when a large deposit occurred, because an early deposit will have more time to accrue interest than a late deposit).

We used the "optimize" function in R to find the value of $r_m$ that enabled us to match the observed number of gorillas at the end of the study period. The monthly growth rate was converted into an annual growth rate ($r_a$) using Eq 3 to account for monthly compounding.

$$(1 + r_a) = (1 + r_m)^{12} \qquad (3)$$

For example, if the monthly growth rate ($r_m$) was 0.1%, then $(1 + r_a)$ would equal $(1 + 0.001)^{12}$ which equals 1.012. The annual growth rate ($r_a$) would equal 0.012 or 1.2% per year. We ran 100 iterations of the growth rate calculations, in which we assumed that all of the unexplained disappearances had the same randomly generated probability of being caused by death rather than dispersal ("death probability"). In each iteration, we then applied separate random numbers (between 0–1) to each unexplained disappearance. When the random number was less than the death probability, we assumed that the gorilla had died, otherwise the disappearance was counted as outflow in Eq 2. We ran 100 sets of those 100 iterations, which each used a separate death probability. Thus, we ran a total of 10,000 iterations which controlled for uncertainty in the death probability, as well as the variability in the growth rate estimates at a given death probability. We report the average growth rate and the 95% confidence limits from those 10,000 iterations.

## Density dependence

To test for the effects of density dependence upon offspring mortality, we ran a generalized linear mixed model (GLMM) with one data point for each offspring. The response variable

equaled "1" if the offspring died before reaching age four (the typical weaning age), and "0" if it did not. The predictor variable was the study population size when the offspring was born. By using the study population size as a proxy for the gorilla density, we were essentially assuming that those two variables were positively correlated. Random effect variables were the identity of the mother and the group where the offspring was born. The model excluded offspring that were born during the last four years of observations, because we could not always determine whether those infants would survive to reach age four.

To test for density dependence upon the age of first parturition, we ran a GLMM with one data point for each female whose first birth was observed. The response variable equaled the age of the mother. The predictor variable was the study population size when the offspring was born. Random effect variables were the identity of the mother and the group where the offspring was born. We excluded females whose birth date and and/or first birth was not known to within three months.

To test for density dependence upon female fertility, we ran a GLMM with one data point for each interbirth interval. The response variable equaled the length of the interbirth interval. The predictor variable was the study population size when the offspring was born. Random effect variables were the identity of the mother and the group where the offspring was born. Interbirth intervals can be significantly shorter when an infant dies, so the analyses were limited to offspring that survived to reach age four. We excluded intervals that began/ended with a birth whose date was not known to within three months. We also excluded intervals when the mother temporarily left the study groups, because we could have missed infants who died shortly after birth.

It is not visually apparent when female gorillas are pregnant, so infants can go undetected if they die shortly after birth, even in habituated groups that are monitored every day [70]. Therefore, this study has probably underestimated the number of births and infant mortality, while overestimating the length of interbirth intervals. To compensate for this bias, we performed a regression in which the response variable was the rate of giving birth to offspring that survived at least four years. The model had one data point for each year of the study, except for the last four years of observations, because we could not always determine whether those infants would survive to reach age one. The surviving birth rate equaled the number of births of gorillas who survived to reach age four, divided by the number of adult female-years. We calculated the adult female-years by adding the number of females at the beginning of each month (i.e., female months), and then dividing by twelve. The predictor variable was the study population size at the beginning of the year. To avoid excessive influence from years with less data, each data point was weighted according to the number of female-years observed.

To test for density dependence on adult female mortality and/or dispersal out of the study area, we ran a linear regression with one data point for each year. The response variable was the rate of unexplained disappearances, which equaled the number of unexplained disappearances that occurred during the year, divided by the number of female-years that were observed. The predictor variable was the study population size at the beginning of the year. To avoid excessive influence from years with less data, each data point was weighted according to the number of female-years observed.

To test for the density dependence upon dominant male mortality, we ran a GLMM with one data point for each month that each dominant male was observed. The response variable equaled "1" if the dominant male survived the month, and "0" if he disappeared. The predictor variable was the number of study groups at the beginning of the month. Thus, we were examining the potential effects of group density (rather than population density) to provide a consistent basis for comparison with the previous study of the Virunga mountain gorillas [27]. The age of the dominant male was included as a control variable, and the identity of the dominant male was a random effect variable.

To test for density dependence in the rate of suspected infanticide, we ran a GLMM with one data point for each offspring. The response variable equaled "1" if the infant disappeared during a group disintegration, and "0" if it did not. The predictor variable was the number of study groups when the offspring was born. Random effect variables were the identity of the mother and the dominant male when the offspring was born. The age of the dominant male was included as a control variable. The model excluded offspring that were born during the last three years of observations. The model did not account for any infanticide that might occur while the dominant male was alive, but those cases would be included in the model for overall infant mortality.

## Social structure and age structure

To examine the overall variance in group size, we ran an ANOVA with one data point for each group on the first day of each year. The response variable was the average group size, and the predictor was a category variable for the year. The $R^2$ from the ANOVA indicates the proportion of overall variance that occurred among years, versus the variance among groups within each year. We also ran a t-test which compared the group size at the beginning of the study (January 1996) versus the end of the study (May 2020). The t-test had one data point for each group in each of those two months.

To examine the potential influence of population size upon social structure, we ran Spearman correlations with one data point for the first day of each year. One test examined the correlation between the population size and the number of groups in the study. The other test examined the correlation between the population size and the average group size.

## Permission and/or consent for the study to take place

The Ministry of Sustainable Development, Forest Economy and Environment and the Ministry of Scientific Research of the Republic of Congo provided permission to conduct field work in the Nouabalé Ndoki National Park.

## Ethical statement

For this study, gorillas were observed in their natural habitat. Our observations were made from a 9-m-high platform and animals were mostly unaware of researchers' presence; we therefore believe that our study had very little disturbance to the animals or the ecosystem. All study procedures complied with the Comité d'Ethique de la Recherche en Sciences de la Santé in Brazzaville, Republique of Congo, the ethical standards of the Max Planck Institute for Evolutionary Anthropology, and the primatology department's ethical guidelines for non-invasive research.

## Results

### Population increase

The study began with 101 gorillas on January 1, 1996, which represents 19.6% of the 514 gorillas that were ultimately identified (Table 1). Another 160 gorillas subsequently joined the study population from the surrounding area (31.1%), and 253 gorillas were born during the study (49.2%). There were 266 direct transfers between social units within the population, and 38 temporary disappearances (Table 2). The median length of the temporary disappearances was 310 days (InterQuartile Range = 185–550 days). The study had 137 permanent unexplained disappearances (26.7% of the 514 gorillas), 151 assumed deaths (29.4%), and it ended with 226 gorillas on May 1, 2020 (44.0%).

**Table 1. Sample sizes for the initial status and final fate of each gorilla in the study.**

| Class | start of study | births | influx | end of study | deaths | unex | gorilla years |
|---|---|---|---|---|---|---|---|
| Infants | 22 | 253 | 3 | 34 | 109 | 0 | 734.7 |
| Juveniles | 8 | 0 | 16 | 33 | 18 | 5 | 423.9 |
| Subadults | 11 | 0 | 36 | 15 | 5 | 16 | 310.2 |
| Unknown | 0 | 0 | 0 | 4 | 0 | 0 | 7.3 |
| Adult females | 34 | 0 | 62 | 73 | 0 | 73 | 1338.0 |
| Blackbacks | 7 | 0 | 16 | 7 | 0 | 7 | 193.8 |
| Young silverbacks | 1 | 0 | 12 | 9 | 0 | 19 | 240.4 |
| Adult silverbacks | 18 | 0 | 15 | 51 | 19 | 17 | 653.3 |
| Total | 101 | 253 | 160 | 226 | 151 | 137 | 3901.5 |

The initial status includes gorillas who were already identified at the beginning of 1996 ("start of study"), or were born during the study ("births"), or were counted as influx. The final fate includes gorillas who were still observed as of May 2020 ("end of study"), or they are assumed to have died ("deaths"), or they are considered unexplained disappearances ("unex"). See Methods for more details about each type of initial status, final fate, and age/sex category. Gorillas in the "unknown" age/sex category were approximately twelve years old, so they could have been blackbacks or nulliparous adult females.

Movements include the number of gorillas who initially appeared through locational immigration or social immigration, as well as internal transfers, temporary disappearances, and reappearances, and some of the unexplained disappearances (Table 1). All reappearances occurred through social immigration into a known group, so they are included in the tally for social immigration. See Methods for more details about each type of movement and the age/sex categories.

The influx of 160 gorillas was comprised of social dispersal by 116 gorillas into study groups that were already known (72.5%), and the locational dispersal of 44 gorillas in previously unknown social units that entered the study area (27.5%). The social dispersal included the immigration of 33 juveniles, subadult males, and blackbacks into study groups. Social dispersal in those age/sex categories typically occurs when a group disintegrates after the death of the dominant male, yet only twelve gorillas in those categories disappeared from the study population during a group disintegration. Even if all of those disappearances were outflow, the categories would have a 33:12 ratio of influx:outflow, which suggests that the surrounding areas had more disintegrations than the study population. The surrounding areas may also have more locational dispersal, because six groups joined the study population while none left. A

**Table 2. Sample sizes for the number of gorilla movements and gorilla-years.**

| class | Locational immigration | Social immigration | Internal transfers | Temporary disappearances | Re-appearances |
|---|---|---|---|---|---|
| infants | 2 | 1 | 6 | 1 | 0 |
| juveniles | 3 | 13 | 17 | 1 | 2 |
| subadults | 3 | 33 | 40 | 10 | 2 |
| unknown | 0 | 0 | 0 | 0 | 0 |
| adult females | 10 | 52 | 115 | 24 | 30 |
| blackbacks | 4 | 12 | 19 | 2 | 4 |
| young silverbacks | 7 | 5 | 59 | 0 | 0 |
| adult silverbacks | 15 | 0 | 10 | 0 | 0 |
| total | 44 | 116 | 266 | 38 | 38 |

net influx also occurred with subadult females, who had 22 appearances through social immigration, versus only eight unexplained disappearances. Thus, at least three separate mechanisms showed a net influx of gorillas into the study: involuntary dispersal after group disintegrations, locational dispersal of groups, and voluntary transfers by subadult females. For two other mechanisms, the locational dispersal by solitary males and voluntary transfers by adult females, the net balance depends on how many unexplained disappearances were due to death versus dispersal (See Supporting Information). Overall, the study population would have a net influx of 23 gorillas, even in the unlikely event that all of the unexplained disappearances had been dispersal to the surrounding areas (rather than death).

The overall increase in the study population from 101 to 226 gorillas represents an annual rate of 3.4%, which reflects both the net influx and the inherent growth rate (Fig 1). After adjusting for the net influx of gorillas, the inherent growth rate had an average value of 0.7% per year, with 95% confidence limits between -0.7% and 2.6%. The confidence intervals represent the uncertainty in the fate of unexplained disappearances.

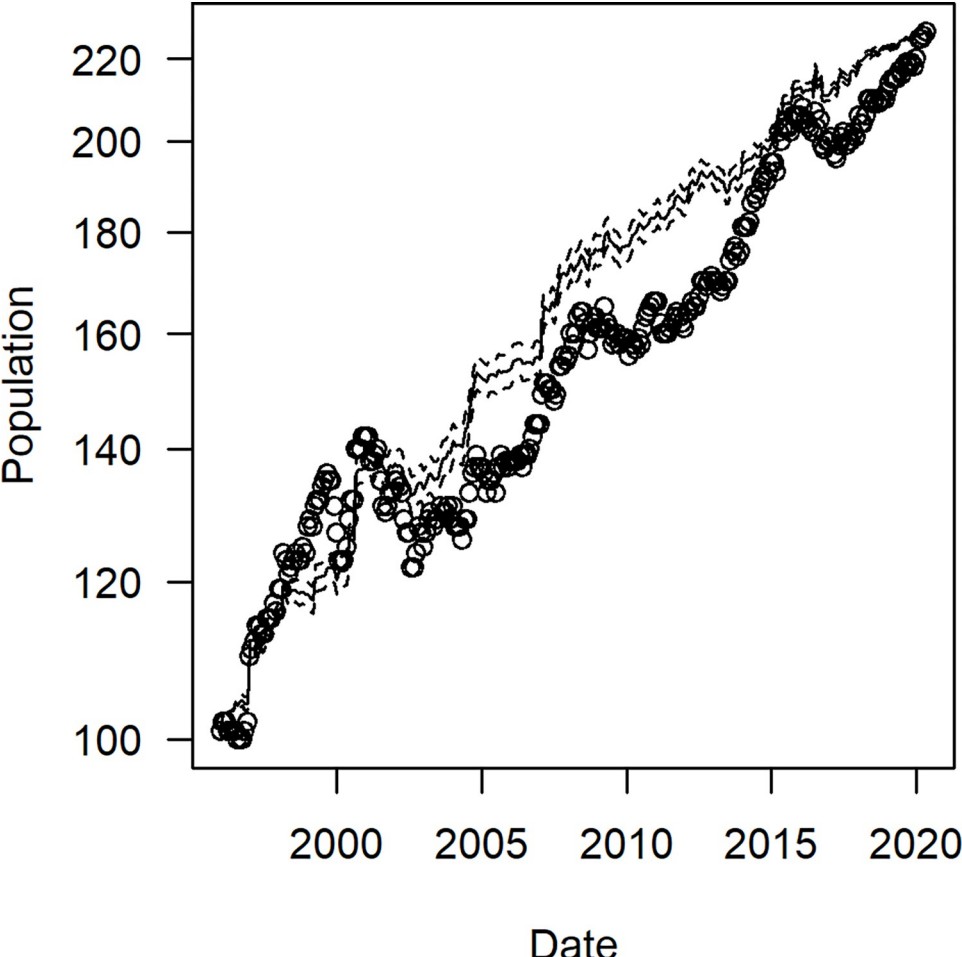

**Fig 1. Temporal variations in the size of the study population.** The solid line is the prediction from a linear regression and the dashed lines show the 95% prediction intervals. The y-axis has a log-scale so a constant rate of increase appears as a straight line. The slope of the line indicates that the population increased at an average annual rate of 2.9%, versus the estimate of 3.4% that was based solely on the initial and final sizes.

## Density dependence

Our proxy variable for gorilla density (the number of gorillas in the study population) was not significantly correlated with the age of first birth, female fertility, nor adult female deaths and outflow (Table 3). Infant mortality was negatively correlated with population density, which is in the opposite direction of expectations. Thus, we found no significant decline in female reproductive success, even as the size of the study population doubled. After excluding the last four years of the study, 96 of 211 infants died before reaching age four (45%). The age of first birth occurred at an average of 12.2 ± 0.6 SD years (N = 7 females), and the length of interbirth intervals was 65.3 ± 10.2 months (N = 48 intervals). After excluding the last four years of the study, the dataset contained 1017.9 female-years and 114 births of gorillas that survived to reach age four, which corresponds to a surviving birth rate of 0.112 births per female-year. In the full dataset, there were 73 permanent disappearances of adult female western gorillas in 1302.4 female-years, which corresponds to a rate of 0.056 deaths and outflow per female-year.

Our proxy variable for group density (the number of study groups) was not significantly correlated with the probability that a dominant male would die in a given month, nor with the probability that an infant would die from the suspected infanticide that occurs when a group disintegrates (Table 3). Thus, we found no significant evidence of density dependence upon male mating competition. There were 19 disappearances of dominant males during the 437.1 dominant male-years observed, which corresponds to a mortality rate of 0.043 deaths per male-year. After those dominant males died, nine infants disappeared, which are considered cases of suspected infanticide as the group disintegrated. Those deaths correspond to 4.3% of the 211 infants that were born during the study.

## Social structure and age structure

The initial population contained 11 breeding groups, two nonbreeding groups, and five solitary males. The number of breeding groups increased from 11 to 21 during the study (Fig 2). The average size of breeding groups was 8.1 ± 2.2 gorillas at the beginning of the study, which is not significantly different from 8.0 ± 4.1 at the end of the study (t-test: t = 0.14, df = 10.7, p = 0.89). The increase in population size was more strongly correlated with the number of breeding groups (rho = 0.92) than with the average size of those groups (rho = -0.33). Based on a snapshot of the population on the first day of each year, only 3% of the variance in breeding group size occurred among years, with the remaining variance arising among breeding groups on each day. Thus, the population had a fairly stable social structure throughout the study.

The proportion of immature gorillas in study groups averaged 41.0% ± 3.1%, which is slightly above the 40% threshold that has been considered evidence of a stable or growing

**Table 3. Details from statistical models for density dependence.**

| Model | Predictor | Estimate | StdErr | t/z value | p-value |
|---|---|---|---|---|---|
| infant mortality | gorillas | -0.40256 | 0.18246 | -2.21 | 0.021 |
| age of first birth | gorillas | 0.03832 | 0.26575 | 0.14 | 0.891 |
| interbirth intervals | gorillas | -1.49546 | 1.31459 | -1.14 | 0.243 |
| surviving birth rates | gorillas | 0.00031 | 0.00046 | 0.68 | 0.505 |
| disappearances | gorillas | -0.00037 | 0.00029 | -1.29 | 0.210 |
| dominant male deaths | groups | -0.11539 | 0.05176 | -2.23 | 0.153 |
| suspected infanticide | groups | -0.95263 | 0.97610 | -0.98 | 0.329 |

Estimate and standard error of the coefficient for the predictor variable in each model (the number of gorillas or groups in the study). The t-value is presented for linear regressions, and the z-value for GLMM (See Methods).

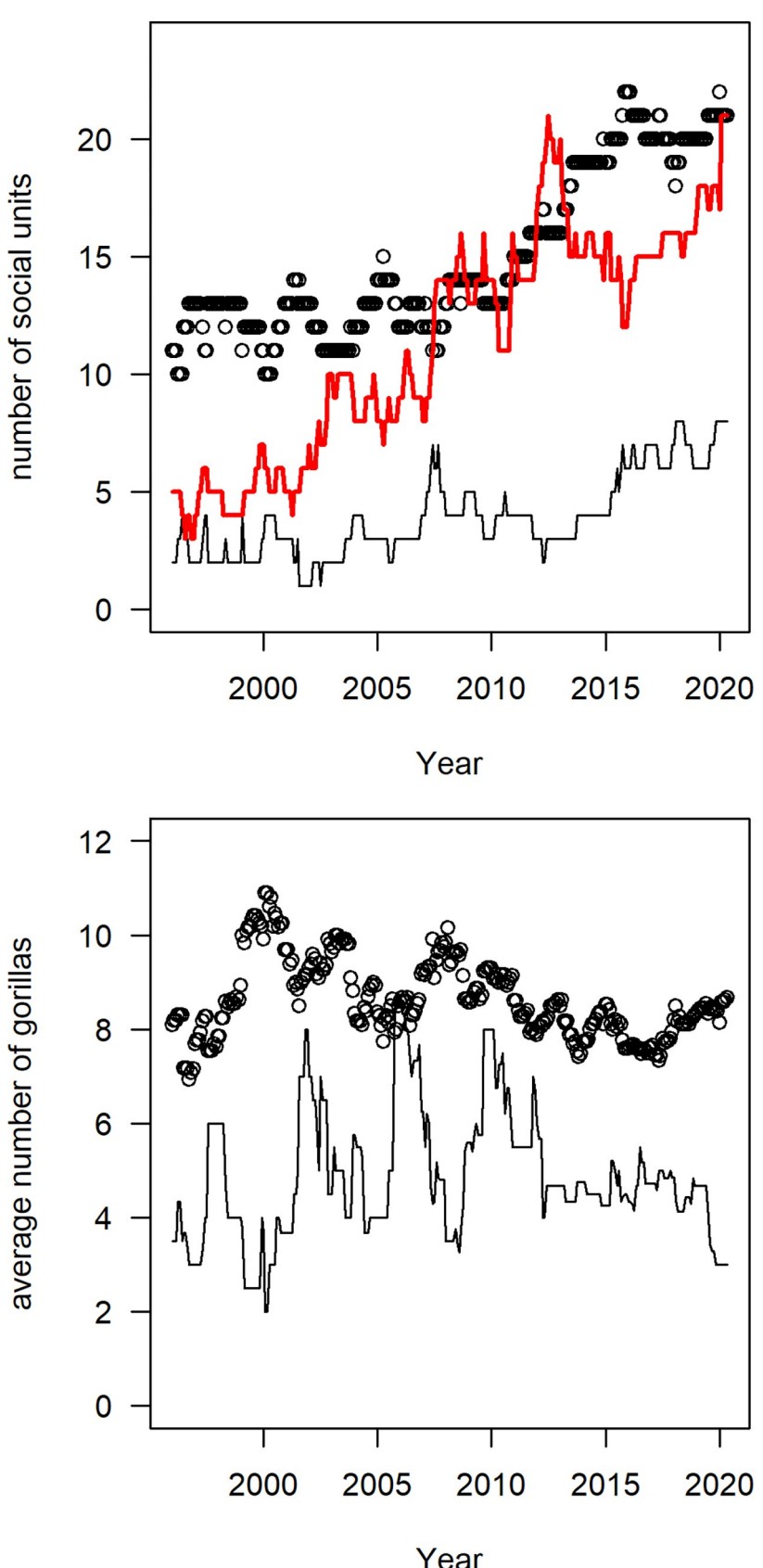

**Fig 2. Temporal variations in the social structure.** Social units include breeding groups (black circles), non-breeding groups (black lines), and solitary males (red lines). The graphs show the number of each type of social unit (a) and the average size of each social unit (b). By definition, the social unit size for each solitary male equals one gorilla.

population for mountain gorillas [62]. The proportion of immatures was 42.7% at the beginning of the study, which is not significantly different from 40.7% at the end of the study (Fisher exact test, p = 0.80). The proportion of infants seemed to fluctuate in intervals of 5–6 years, which may reflect the average interbirth interval between surviving offspring (Fig 3). The oscillations in the proportion of infants did not translate as clearly into the subsequent age classes of juveniles or subadults. The adult sex ratio was 1.9: 1 at the beginning of the study, which is not significantly different than 1.5: 1 at the end of the study (Fisher exact test, p = 0.50). Thus, the population seemed to have a stable age structure throughout the study.

## Discussion

### Population growth

The western gorillas at Mbeli bai had an inherent growth rate of 0.7% per year, with 95% confidence limits between -0.7% and 2.6%. Our methodology does not track the fate of individuals who disappear permanently, so the confidence limits reflect uncertainty about whether those disappearances were due to death or dispersal. The average growth rate suggests that the local population is stable or increasing. Even the lower 95% confidence limit is better than the average annual declines of 2.7% for the overall population of western gorillas [36, 71, 72]. Thus, this study suggests that the establishment of the Nouabalé-Ndoki National Park (NNNP) in 1993, along with conservation measures in surrounding areas, may be preventing the losses that have occurred elsewhere for western gorillas. Other subspecies of gorillas have fared even more poorly. The total population of Grauer's gorillas has declined by more than 50%, and only a few hundred Cross River gorillas remain [73, 74]. The most encouraging trend has been the growth within both populations of mountain gorillas, but extreme conservation efforts have been needed [22, 30, 68, 75]. In addition to differences in human disturbances and disease, the variation in population dynamics among gorillas could reflect differences in life history responses to ecological conditions, as well as differences in population density relative to carrying capacity [65, 76]. See the Supporting Information for more detailed comparisons of western gorillas versus mountain gorillas.

The Mbeli population increased more rapidly than its inherent growth rate, due to a net influx of gorillas from the surrounding areas (even if we assume that all unexplained disappearances were outflow). Thus, the study groups have not been a "source" of gorillas to support the broader population [12, 77]. Long term monitoring at bais has also revealed a net influx of individuals in other species such as elephants and sitatunga [78, 79]. Our results would fit the definition of a "sink" if the inherent growth rate of Mbeli was negative [11, 13]. Mbeli may not quite match the previous examples of a "refuge", however, because the gorilla density seems to be lower than some adjacent areas [15, 41, 80]. Logging in nearby areas may have temporarily displaced gorillas, but it is unknown whether any of them moved to the Mbeli area [40]. We would expect such displacement to involve locational dispersal of entire groups from one area to another, whereas the influx in our study mainly arose from social dispersal of individuals from one group to another. Further investigation of the surrounding areas could enhance our understanding of the source-sink dynamics at Mbeli, and contribute to conservation management decisions about the best places to allocate resources [81, 82].

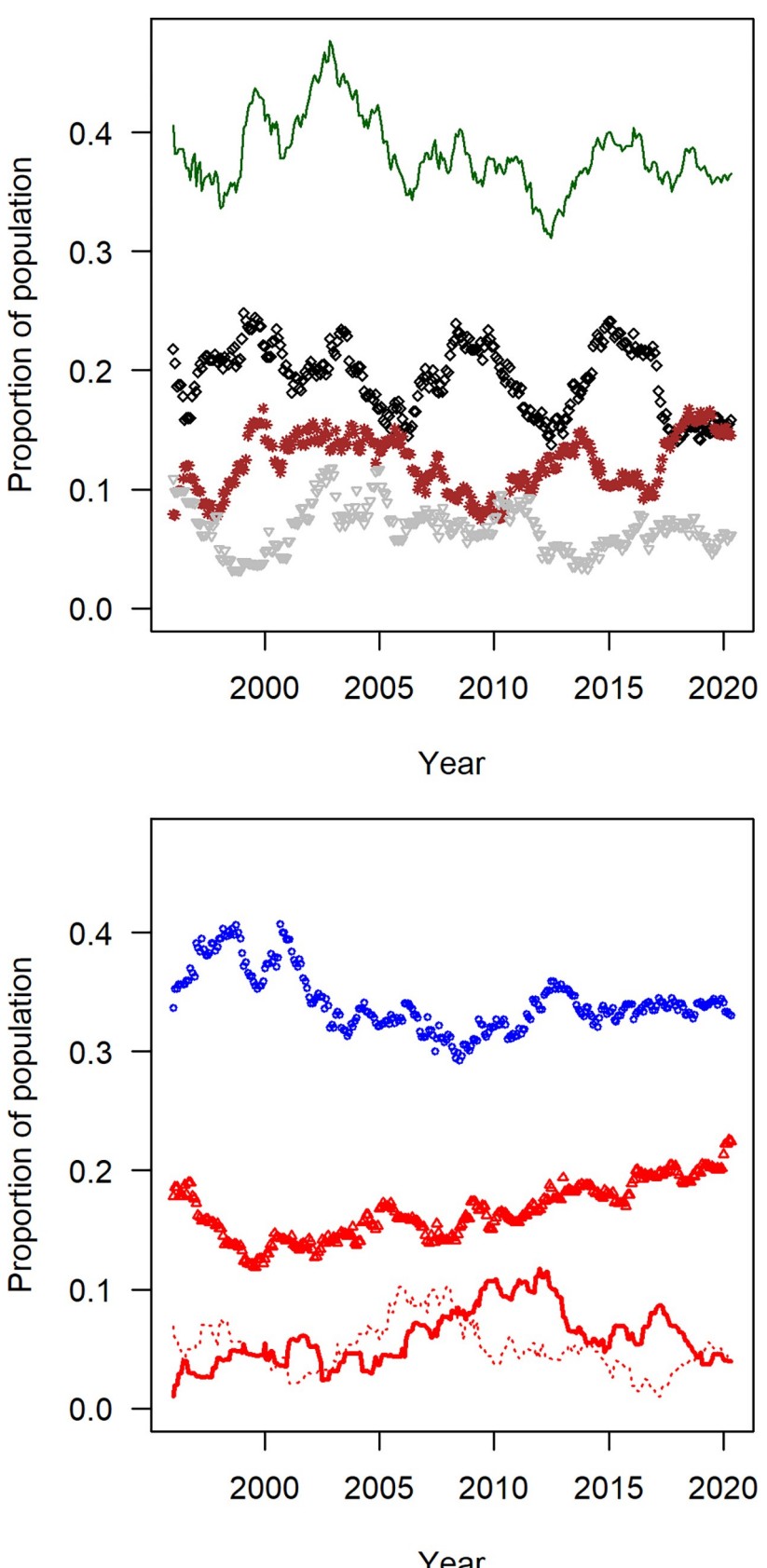

**Fig 3. Proportion of gorillas in each age/sex classification.** Immature gorillas (green line in a) include infants (black diamonds), juveniles (brown asterisks), and subadults (grey inverted triangles). Mature gorillas in b consist of adult females (blue circles), adult silverbacks (red triangles), young silverbacks (thick red line), and backblacks (dotted red line).

## Density dependence

As the population size doubled, our analyses showed no significant decreases in female reproductive success. Those results suggest that the population density is not nearing its carrying capacity. In contrast with our results, increases in population density have led to lower fertility in howler monkeys (*Alouatta seniculus*) and rhesus macaques (*Macaca mulatta*) [83, 84]. Surprisingly, increases in population density led to higher fertility for northern muriquis (*Brachyteles arachnoides*), who essentially increased their carrying capacity by expanding their habitat [5]. Collectively, those studies are consistent with a comparison of mountain gorilla populations, where the potential impact of carrying capacity was more apparent for fertility than infant mortality, as predicted for species that are not annual breeders [22]. Those previous results also resemble the effects of environmental variability, which has shown greater impact on fertility than the survival of adult females [85, 86]. If so, then such results would support expectations that demographic buffering against environmental variability will be greatest for the vital rates that have the greatest influence on population growth [87–89]. Similarities in the effects of density dependence versus environmental variability may arise when they both reflect the same underlying factors, such as changes in the food availability per capita [90].

As the number of groups doubled in this study, we did not see a significant increase in dominant male mortality, nor in the rate of subsequent infanticide that is expected when groups disintegrate. Thus, we found no evidence that the population growth is limited by density dependence in male mating competition [91]. Our results contrast with the Virunga mountain gorillas, where a proportional increase in group density coincided with the death of seven adult males, and a five-fold increase in infanticide [27]. One potential explanation for this discrepancy is that our group density remained lower than the Virungas (See the Supporting Information). Rather than increasing steadily with group density, the frequency of infanticide might not intensify until the population exceeds some threshold. The typical threshold could be even higher than the Virungas, because the results for mountain gorillas reflected exceptional circumstances [27]. The mountain gorilla study began with an extraordinary number of young silverbacks in groups with old dominant males, and the increased aggression may have been catalyzed by unstable agonistic relationships as those young silverbacks became established [92, 93]. If so, then some populations might never reach the threshold, especially if their density is limited by different factors than the Virunga mountain gorillas, who have abundant food, veterinary care, and no natural predators [30]. Thus, density dependence of male mating competition could have a greater impact on the population dynamics of the Virunga mountain gorillas than other species.

## Social structure and age structure

The average group size did not change significantly during the study, so the population growth was mainly reflected in an increase in the number of groups (from 11 to 21 breeding groups). Those results again contrast with the Virunga mountain gorillas, where population growth has coincided with an increase in both group size and the number of groups [68, 69, 94, 95]. Our results are consistent with the possibility that western gorillas face greater socioecological constraints on group size than mountain gorillas [49]. Western gorillas could face greater constraints due to feeding competition, if they have lower food density that leads to greater travel

requirements that cannot be mitigated by increasing group spread [58, 93, 96–98]. Western gorillas could face greater constraints due to male mating competition, if their lack of multi-male groups makes it harder to retain a large number of females [60, 99–101]. In either case, a strongest test for greater socioecological constraints is simply whether western gorillas have smaller groups than mountain gorillas [49, 68, 102, 103]. Such tests can be supported by more detailed evidence of feeding competition and/or male mating competition, such as greater travel requirements for larger groups [58, 104]. Our current results provide further support by illustrating that the smaller groups sizes of western gorillas are not merely an artifact of transient conditions, because they were maintained even as the population size doubled.

The proportion of immature gorillas was above 40%, which has been considered evidence of a stable or growing population for mountain gorillas [61, 62]. The average value of our growth rate estimate may suggest that the 40% threshold can apply to western gorillas, but the lower 95% confidence limit does not exclude the possibility that the study population was declining. Other aspects of the age distribution (and social structure) of western gorillas have been proposed as indicators of Ebola outbreaks, which affect adult females and immature individuals more than solitary males [57, 71]. More broadly, a stable or growing population is often expected to have a high ratio of immatures (or infants) to adult females, which presumably reflects a high rate of giving birth to surviving offspring [105, 106]. Like any proxy, these indicators should be considered tentative because population growth and age structure can depend on a variety of demographic parameters, such as the rates of births, deaths, and dispersal [23, 94, 107].

## Conclusions

The population of western gorillas at Mbeli bai has increased for 25 years while the overall species has declined [36]. We found no significant evidence of density-dependent constraints on carrying capacity, even as the number of study groups doubled. Our results provide evidence of stability or population increase on a relatively small spatial scale for western gorillas, and they highlight the value of law enforcement and protected areas, but they do not diminish the importance of improving conservation for this critically endangered species [36, 108]. Western gorillas have a slow life history, so even with a positive growth rate, a long time is needed to recover from population disturbances [109]. Our results support previous studies showing the value of long-term research sites for conservation [15, 16, 110]. The nearby surrounding areas may be faring differently from Mbeli, so further investigation could help to inform broader conservation efforts [111–113].

## Supporting information

**S1 File. Additional analyses and discussion of the growth rate, influx, and outflow.**
(DOCX)

**S1 Data. Data used for each of the tables, figures, and analyses.**
(RDATA)

## Acknowledgments

We thank the Ministry of Sustainable Development, Forest Economy and Environment and the Ministry of Scientific Research of the Republic of Congo for permission to conduct field work in the Nouabalé Ndoki National Park. We also thank the Nouabalé-Ndoki Foundation and the Wildlife Conservation Society-Congo Program for the administrative and logistical support. We are grateful to the numerous researchers and assistants for their help monitoring

Mbeli Bai over the years. We thank Vittoria Estienne for managerial support and for comments on a previous version of the manuscript. We thank Fernando Colchero for sharing details from his estimated survivorship curves (see Supporting Information).

## Author Contributions

**Conceptualization:** Andrew M. Robbins, Marie L. Manguette, Martha M. Robbins.

**Data curation:** Marie L. Manguette, Thomas Breuer, Milou Groenenberg, Richard J. Parnell, Claudia Stephan, Emma J. Stokes, Martha M. Robbins.

**Formal analysis:** Andrew M. Robbins.

**Investigation:** Andrew M. Robbins, Marie L. Manguette.

**Methodology:** Andrew M. Robbins, Marie L. Manguette.

**Supervision:** Martha M. Robbins.

**Writing – original draft:** Andrew M. Robbins, Marie L. Manguette, Martha M. Robbins.

**Writing – review & editing:** Andrew M. Robbins, Thomas Breuer, Richard J. Parnell, Claudia Stephan, Martha M. Robbins.

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
