## [Decision Letter · Decision Letter 0]

24 Aug 2022

PONE-D-22-19909Population dynamics of western gorillas at Mbeli BaiPLOS ONE

Dear Dr. Robbins,

Thank you for submitting your manuscript to PLOS ONE. After careful consideration, we feel that it has merit but does not fully meet PLOS ONE’s publication criteria as it currently stands. Therefore, we invite you to submit a revised version of the manuscript that addresses the points raised during the review process.

We look forward to receiving your revised manuscript.

Kind regards,

Bi-Song Yue, Ph.D

Academic Editor

PLOS ONE

Journal Requirements:

Reviewers' comments:

Reviewer's Responses to Questions

**Comments to the Author**

1. Is the manuscript technically sound, and do the data support the conclusions?

Reviewer #1: Yes

Reviewer #2: Yes

2. Has the statistical analysis been performed appropriately and rigorously? 

Reviewer #1: Yes

Reviewer #2: Yes

3. Have the authors made all data underlying the findings in their manuscript fully available?

Reviewer #1: Yes

Reviewer #2: Yes

4. Is the manuscript presented in an intelligible fashion and written in standard English?

Reviewer #1: Yes

Reviewer #2: Yes

5. Review Comments to the Author

Reviewer #1: The study is well comprehensive addressing density dependance and population dynamics of western gorillas at Mbeli Bai. I have no corrections for this research paper. The author investigates the population dynamics of western gorillas in Mbeli Bai, a swampy forest clearing within the Nouabalé-Ndoki National Park (NNNP) in the Republic of Congo, over a 25-year period. Despite their widespread range in Central Africa, western gorillas (Gorilla gorilla gorilla) are critically endangered. The entire population of western gorillas is estimated to be 330,000, with a 2.7 percent annual decline owing to illicit murders, sickness, and habitat destruction.

Reviewer #2: The abstract needs some minor corrections.

The introduction is too clumsy, losing clarity and relevance. Some of the observations are out of context and need to be streamlined. The objective of the study was not mentioned properly.

Methods require additional clarification on the growth rate in equation number 3. The authors also missed important variables like ecological and environmental parameters and their impact on the population. Is there any correlation between the emergence of bais and stable population growth?

I am not clear about the percentile figure of gorillas presented in the first paragraph of the result. Is this figure inclusive of perinatal mortality, disease-related deaths, infanticides, and so on?

 Line no 409-411:  Is there any evidence of lower population growth or losses prior to the establishment of the Nouabalé-Ndoki National Park (NNNP) in 1993?

Lines 434-462: The observations are intriguing, and one inference line should be drawn from them at the end of the second paragraph.

6. PLOS authors have the option to publish the peer review history of their article (what does this mean?). If published, this will include your full peer review and any attached files.

Reviewer #1: No

Reviewer #2: No

---

## [Author Response · Author response to Decision Letter 0]

16 Sep 2022

POPULATION DYNAMICS OF WESTERN GORILLAS AT MBELI BAI

REVIEWERS’ COMMENTS:

Reviewer #1: 

The study is well comprehensive addressing density dependance and population dynamics of western gorillas at Mbeli Bai. I have no corrections for this research paper. The author investigates the population dynamics of western gorillas in Mbeli Bai, a swampy forest clearing within the Nouabalé-Ndoki National Park (NNNP) in the Republic of Congo, over a 25-year period. Despite their widespread range in Central Africa, western gorillas (Gorilla gorilla gorilla) are critically endangered. The entire population of western gorillas is estimated to be 330,000, with a 2.7 percent annual decline owing to illicit murders, sickness, and habitat destruction.

>Thank you for the comments.

Reviewer #2: 

The abstract needs some minor corrections.

>We have now reworded some sentences in the Abstract.

The introduction is too clumsy, losing clarity and relevance. Some of the observations are out of context and need to be streamlined. The objective of the study was not mentioned properly.

>To streamline the Introduction, we removed most of the fifth paragraph which included methods that are not relevant to this study. On line 73 of the original pdf, we reworded the sentence to explicitly state the overall objective of the study. More detailed objectives are provided in the last paragraph of the Introduction (e.g.,“to determine whether Mbeli is a source, a sink, or a refuge”, “to look for potential effects of density dependence upon female reproductive success”, etc.).

Methods require additional clarification on the growth rate in equation number 3. The authors also missed important variables like ecological and environmental parameters and their impact on the population. Is there any correlation between the emergence of bais and stable population growth?

>We have now added an example to illustrate the conversion from a monthly growth rate to an annual growth rate using Equation 3. We agree that it would be worthwhile to include any temporal variations in ecological conditions around the bai, but unfortunately such information was not available. We are unaware of any correlation between the emergence of bais and population growth.

I am not clear about the percentile figure of gorillas presented in the first paragraph of the result. Is this figure inclusive of perinatal mortality, disease-related deaths, infanticides, and so on?

>After line 180 of the original pdf, we have now stated that the assumed deaths could be due to predation, disease, infanticides, or other causes. The study was unlikely to detect perinatal mortality because the gorillas were not observed daily, and it is not visually apparent when they are pregnant. 

Line no 409-411: Is there any evidence of lower population growth or losses prior to the establishment of the Nouabalé-Ndoki National Park (NNNP) in 1993?

>We are unaware of any studies of population changes before 1993.

Lines 434-462: The observations are intriguing, and one inference line should be drawn from them at the end of the second paragraph.

>On line 462, we have now stated that density dependence of male mating competition could have a greater impact on the population dynamics of the Virunga mountain gorillas than other species.

Editor

>Sorry, but we couldn’t see how our manuscript differs from the style requirements in those links. Could you please be more specific about any changes that are needed?

2. In your Data Availability statement, you have not specified where the minimal data set underlying the results described in your manuscript can be found. PLOS defines a study's minimal data set as the underlying data used to reach the conclusions drawn in the manuscript and any additional data required to replicate the reported study findings in their entirety. All PLOS journals require that the minimal data set be made fully available. >We revised the Data Availability statement to indicate that the data is in a Supporting Information file. We named the file “ArchiveData_PopulationDynamics_WesternGorillas”, but the name somehow got encrypted when we uploaded it, which might make it harder to find.

---

## [Editor Report · Decision Letter 1]

20 Sep 2022

Population dynamics of western gorillas at Mbeli Bai

PONE-D-22-19909R1

Dear Dr. Robbins,

We’re pleased to inform you that your manuscript has been judged scientifically suitable for publication and will be formally accepted for publication once it meets all outstanding technical requirements.

Kind regards,

Bi-Song Yue, Ph.D

Academic Editor

PLOS ONE

---

## [Editor Report · Acceptance letter]

26 Sep 2022

PONE-D-22-19909R1 

Population dynamics of western gorillas at Mbeli Bai 

Dear Dr. Robbins:

I'm pleased to inform you that your manuscript has been deemed suitable for publication in PLOS ONE. Congratulations! Your manuscript is now with our production department. 

Kind regards, 

on behalf of

Dr. Bi-Song Yue 

Academic Editor

PLOS ONE